# Impact of Anti-IL5 Therapies on Patients with Severe Uncontrolled Asthma and Possible Predictive Biomarkers of Response: A Real-Life Study

**DOI:** 10.3390/ijms24032011

**Published:** 2023-01-19

**Authors:** Susana Rojo-Tolosa, María Victoria González-Gutiérrez, Gonzalo Jiménez-Gálvez, José Antonio Sánchez-Martínez, Laura Elena Pineda-Lancheros, José María Gálvez-Navas, Alberto Jiménez-Morales, Cristina Pérez-Ramírez, Concepción Morales-García

**Affiliations:** 1Respiratory Medicine Department, University Hospital Virgen de las Nieves, 18014 Granada, Spain; 2Pharmacy Service, Pharmacogenetics Unit, University Hospital Virgen de las Nieves, 18014 Granada, Spain; 3Center of Biomedical Research, Department of Biochemistry and Molecular Biology II, Institute of Nutrition and Food Technology “José Mataix”, University of Granada, Avda. del Conocimiento s/n., 18016 Granada, Spain

**Keywords:** severe uncontrolled asthma, mepolizumab, benralizumab, effectiveness, biomarkers

## Abstract

Severe Uncontrolled Asthma (SUA) counts for more than 25% of cases of severe asthma. The main factors that impair the quality of life of these patients are high doses of oral corticosteroids, the presence of exacerbations, and reduced lung function. The objective of this study was to evaluate, in real life, the clinical improvement of patients with SUA treated with anti-interleukin 5 (IL5) therapies: mepolizumab and benralizumab, together with the search for biomarkers associated with the response. We conducted a retrospective observational cohort study that included patients with severe uncontrolled eosinophilic asthma in a tertiary hospital receiving biological therapies. Three types of response were evaluated: improvement in lung function, reduction in exacerbations, and decrease in the use of oral corticosteroids. After 12 months of treatment, significant reductions were found in the number of exacerbations, the use of oral corticosteroids, and blood eosinophil levels for both biological therapies (*p* < 0.001). Lung function improved, achieving a significant improvement in %FEV1 (*p* < 0.001), as well as asthma control, with a significant increase in asthma control test (ACT) scores in both therapies. The markers associated with the corticosteroid-saving effect were the low doses of oral corticosteroids and absence of exacerbations for mepolizumab, and higher blood eosinophilia, absence of chronic obstructive pulmonary disease (COPD), and reduction in oral corticosteroid cycles for benralizumab. The greatest improvement in lung function in both therapies was linked to lower previous FEV1 levels and absence of other respiratory diseases. The reduction in exacerbations was associated with absence of exacerbations the previous year for mepolizumab and never smokers for benralizumab. The results of this real-life study confirm the clinical benefit obtained after the introduction of an anti-IL5 biological therapy and the possible predictive biomarkers of response to treatment.

## 1. Introduction

Asthma is a chronic and heterogeneous inflammatory disease of the airways, in which various cells and inflammatory mediators are involved, partly determined by genetic factors, resulting in bronchial hyperreactivity and restriction of airflow, which give rise to recurrent episodes of wheezing, dyspnea, and exacerbations [1,2].

It is one of the main non-transmissible diseases affecting children and adults. The most recent World Health Organization (WHO) estimates indicate that in 2019, 262 million people were affected by asthma worldwide and 461,000 deaths due to lack of diagnosis and/or treatment were recorded that year [3]. As it is a chronic condition, the objective pursued in addressing it is to achieve control of the disease and prevention of future risk, since it can be life-threatening and generates significant welfare costs [2].

According to the European Respiratory Society (ERS) and the American Thoracic Society (ATS), severe uncontrolled asthma (SUA) represents between 5% and 10% of asthma cases, making it a constant challenge to maintain this control [4]. Data reported by the ARS/ATS, along with European studies from recent years, show that more than 50% of asthma patients are uncontrolled [5].

Between 30% and 40% of patients with severe asthma also require regular use of oral corticosteroids (OCS) to control their asthma [6,7,8], which carries the risk of presenting with serious and often irreversible adverse effects [9,10].

The Global Strategy for the Management and Prevention of Asthma (GINA) defines asthmatic exacerbation as episodes characterized by a progressive increase in symptoms of shortness of breath, cough, wheezing or chest tightness, and a progressive decrease in lung function, i.e., they represent a change from the patient’s usual status that is sufficient to require a change in treatment [2]. Exacerbations differ from the usual symptoms of asthma in the lack of response to the inhaled corticosteroids (ICS) in common use and are strongly related to increased inflammation of the airways. They can trigger fatal results and are more common and more severe when the asthma is uncontrolled [11]. They are often related to eosinophilic airway inflammation since the markers of eosinophilic inflammation are increased prior to the onset of exacerbations [12,13,14]. Furthermore, strategies for the control of eosinophilic airway inflammation are associated with a reduction of exacerbations [15,16,17].

Lack of prevention and/or treatment of exacerbations, together with the presence of potentiating risk factors, such as deficient treatment adherence, incorrect inhaler technique, comorbidities, exposure to toxins or allergens, and reduced lung function, represent a major problem in the clinical management of asthma, associated with increased morbidity and risk of death, which could as much as triple healthcare costs due to increased medication requirements, need for visits to emergency departments, and in many cases, hospitalization [2,18,19,20,21,22].

The development of new anti-interleukin-5 (IL5) biological therapies, such as mepolizumab and benralizumab, for managing asthma represents a major advance in addressing it. Mepolizumab and benralizumab are humanized monoclonal antibodies, which respectively act against IL5 and its receptor (IL5R), selectively and effectively inhibiting the eosinophilic cascade [23,24,25,26,27]. These biological therapies have been shown to be effective in randomized controlled trials (RCTs). However, it is known that only a minority of patients with severe asthma would meet the inclusion criteria of a RCT [28]. Consequently, doubts have been raised about the use of these results in clinical practice, and real-life studies have become more influential in supporting the results of RCTs. There are previous efficacy studies demonstrating a significant reduction in blood eosinophil count and exacerbation rate with mepolizumab and benralizumab [29,30,31,32,33,34,35,36,37]. These studies include diverse groups of participants, so precise pooled estimates are needed to assess efficacy and determine biomarker predictors of response in specific populations. The aim of the study was to evaluate the effectiveness of treatment with mepolizumab and benralizumab and to identify markers predictive of response in Caucasian patients from southern Spain.

## 2. Results

### 2.1. Characteristics of the Patients

#### 2.1.1. Characteristics of Patients Treated with Mepolizumab

A total of 89 patients receiving mepolizumab treatment were included in the study. Their clinical and demographic characteristics are shown in Table 1. The mean recorded age was 55.8 ± 13.1 years, with 65.2% women (58/89). The median time in treatment with mepolizumab was 2 [1–4] years, a change to another monoclonal antibody therapy was recorded in 22.2% (18/89), and 74.2% (66/89) had had no previous biological therapies. Most of the patients were overweight or obese: 42.7% (38/89) and 31.5% (28/89), respectively. Only one case of an active smoker was recorded, representing 1.1%. There were 42.7% (38/89) who had had some previous respiratory disease, 42.7% (38/89) with nasal polyps, 49.4% (44/89) with allergies, 39.3% (35/89) with gastroesophageal reflux disease (GERD), 18% (16/89) with sleep apnea-hypopnea syndrome (SAHS), and 15.7% (14/89) with chronic obstructive pulmonary disease (COPD).

During the previous year, all the patients had required ICS, with a median of 500 [500–1000] µg/day, 73.9% (65/89) needed bursts of OCS, with a median of 2 [0–4] cycles, and 6.7% (6/89) required maintenance OCS. Mean %FEV1 was 71.1 ± 23.7, a median baseline blood eosinophil count of 655 [330–905] cells/μL was recorded, and a baseline IgE of 122.55 [30.25–179.75] IU/mL. There were 60.2% (53/89) who suffered at least one exacerbation during the year before the treatment and the median ACT score was 12.5 [9–15.8] points. 

#### 2.1.2. Characteristics of Patients Treated with Benralizumab

Table 2 describes the demographic and clinical characteristics of the 57 patients treated with benralizumab. Their mean age was 58.4 ± 4.3 years, with 64.9% women (37/57), a median duration of the disease of 7 [4–9] years, and a median of 2 [1–3] years in treatment with benralizumab. They had a low rate of change of biological therapy, 8.8% (5/57), and 66.7% (38/57) had had some previous biological therapy. Most of the patients were overweight or obese: 33.3% (19/57) and 45.6% (26/57), respectively, 24.7% (14/57) were former smokers, and there was only one active smoker (1.7%). A total of 45.6% (26/57) had had some previous respiratory disease, 38.6% (22/57) had nasal polyps, 61.4% (35/57) allergies, 40.4% (23/57) GERD, 17.5% (10/57) SAHS, and 19.3% (11/57) COPD.

During the year before treatment with benralizumab, all patients had received ICS, with a median of 1000 [500–1000] µg/day, 87.7% (50/57) had required bursts of OCS, with a median of 2 [1–4] cycles, and 8.8% (5/57) required maintenance OCS. Mean %FEV1 was 71.7 ± 22.52, a median baseline blood eosinophil count of 410 [230–570] cells/μL was recorded, and a baseline IgE of 137.7 [50–758] IU/mL. There were 45.6% (26/57) who suffered at least one exacerbation during the year before the treatment and the median ACT score was 13 [9.5–15.5] points. 

### 2.2. Comparison of Clinical Variables before and after Treatment

#### 2.2.1. Daily Dose of ICS

After 12 months of biological therapy, daily doses of ICS showed significant reductions of 50% for the benralizumab group (*p* = 0.039). No significant differences between the baseline and post-treatment doses were found in the mepolizumab group. The results of the comparative analysis are shown in Table 3.

#### 2.2.2. Bursts of OCS per Year and Maintenance OCS

During the year prior to starting biological therapy, a high use of bursts of OCS was recorded among the patients in the study: 73.9% of the mepolizumab patients and 87.7% of those with benralizumab needed one or more cycles of OCS. After a year with mepolizumab, the use of bursts of OCS had fallen significantly, by 38.5% (*p* < 0.032, Table 3, Figure 1), as had the median doses received per year (*p* < 0.001, Table 3, Figure 1). Patients being treated with benralizumab recorded a significant reduction of 54% in the requirement for bursts of OCS and the median doses of bursts of OCS per year also dropped significantly (*p* < 0.001, Table 3, Figure 1).

Prior to biological therapy, 6.7% of mepolizumab patients and 8.8% of benralizumab patients required maintenance OCS. After 1 year of mepolizumab treatment, the patients requiring maintenance OCS was significantly reduced to 3.8% (*p* < 0.001, Table 3, Image 1). In contrast, maintenance OCS requirements in the benralizumab group of patients, after 1 year of treatment, increased to 14% (*p* = 0.734, Table 3, Figure 1). There were no significant changes in maintenance OCS doses in either treatment group after 1 year.

#### 2.2.3. Lung Function

Lung function showed a significant improvement after 12 months of biological therapy (Figure 2). In mepolizumab, an increase of 10.3% was observed in mean FEV1 (*p* < 0.001, Table 3) and 11.4% in benralizumab (*p* < 0.001, Table 3). Conversely, there were no statistically significant differences in the group of patients with FEV1 values greater than 80% for either of the two biological therapies.

#### 2.2.4. Asthma Control Test (ACT)

The changes in asthma control test scores were statistically significant in all cases. The patients showed an increase of 9.5 points after administration of mepolizumab (12.5 to 22; *p* < 0.001, Table 3) and of 9 points with benralizumab (13 to 22; *p* < 0.001, Table 3).

#### 2.2.5. Frequency of Severe Exacerbations

The proportion of patients with exacerbations fell from 62.2% in the year prior to starting biological therapy to 14.6% with mepolizumab (*p* = 0.001, Table 3, Figure 3) and from 45.6% to 10.5% with benralizumab (*p* < 0.001, Table 3, Figure 3). A significant reduction occurred in the median number of exacerbations for both treatment groups (*p* < 0.001, Table 3).

#### 2.2.6. Inflammatory Markers

Both biological therapies were associated with significant reductions in the blood eosinophil count (Figure 4). The median blood eosinophil count was reduced by 88% in the mepolizumab group (*p* < 0.001, Table 3) and by 98% in the benralizumab group (*p* < 0.001, Table 3).

### 2.3. Clinical Effectiveness

#### 2.3.1. Clinical Effectiveness of Mepolizumab

After 12 months of treatment with mepolizumab, a reduction of 50% or more in OCS bursts was achieved in 44.9% (40/89) of the patients and 22.5% (20/89) did not require cycles of OCS. In addition, 93.3% (83/89) did not require maintenance OCS and 4.5% (4/89) achieved a reduction of 50% or more in the maintenance dose. A total of 48.1% (38/79) of the patients responded satisfactorily, with an increase of at least 10% in FEV1, and 67.9% (55/81) obtained FEV1 values higher than 80%. The rate of exacerbations was reduced by at least 50% in 47.2% (42/89) of cases and 41.6% (37/89) suffered no exacerbation during the follow-up period. The results are shown in Table 4.

#### 2.3.2. Clinical Effectiveness of Benralizumab

Use of benralizumab for 12 months resulted in a reduction in OCS bursts of 50% or more in 50.9% (29/57) of cases and 10.5% (6/57) did not need cycles of OCS. In addition, 85.9% (49/57) did not require maintenance OCS and 5.3% (3/57) achieved a reduction of 50% or more in the maintenance dose. FEV1 (%) increased by at least 10% in 52.1% (25/48) of the patients and 71.4% (35/49) achieved values of more than 80%. The rate of exacerbations showed a reduction of at least 50% in 42.1% (24/57) of the patients and 52.6% suffered no exacerbation requiring emergency department treatment and/or hospitalization. The results are shown in Table 5.

### 2.4. Predictors of Response at 12 Months

#### 2.4.1. Predictors of Response at 12 Months with Mepolizumab

##### Response to Reduction of Oral Corticosteroids Bursts (OCS)

The bivariate analysis showed a satisfactory response associated with lower initial ICS values, absence of OCS bursts during the year prior to mepolizumab, and lower number or absence of exacerbations (the values are shown in detail in Appendix A). After the multivariate analysis was performed, it was found that lower initial ICS values (OR = 0.99; 95% CI = 0.99–0.99) and fewer exacerbations in the previous year (OR = 0.47; 95% CI = 0.33–0.70) were significantly associated with a better therapeutic response. The results of the multivariate analysis are shown in Table 6.

##### Response to Reduction in Oral Corticosteroid Maintenance (OCS)

The bivariate analysis showed a satisfactory response associated with lower baseline OCS doses of maintenance or absence of maintenance OCS, previous FEV1 values <80%, and fewer exacerbations in the previous year (values are shown in detail in Appendix A). Multivariate analysis showed that lower doses of maintenance OCS indicated a better therapeutic response (OR = 0.73; 95% CI = 0.55–0.91; Table 6).

##### Response to Improvement in Lung Function (FEV1)

In the bivariate analysis, a greater improvement in lung function was found in patients who were older at the start of biological therapy, with no SAHS and lower initial FEV1 values (the values are shown in detail in Appendix A). In the multivariate analysis a significant association was found with the absence of SAHS (OR = 14.44; 95% CI = 2.05–146.15; Table 6) and lower initial FEV1 values (OR = 0.89; 95% CI = 0.84–0.93; Table 6).

##### Response to Reduction of Exacerbations

In the bivariate analysis, a greater reduction of exacerbations was associated with the absence of exacerbations in the year prior to the biological therapy and patients who had not received previous biological therapies (the values are shown in detail in Appendix A). In the multivariate analysis, the response was greater in patients who had had no exacerbations in the year prior to the biological therapy (OR = 0.14; 95% CI = 0.01–0.82; Table 6).

#### 2.4.2. Predictors of Response at 12 Months with Benralizumab

##### Response to Reduction of Oral Corticosteroids Bursts (OCS)

In the bivariate analysis, a satisfactory reduction of bursts of OCS was related to absence of COPD, fewer OCS bursts in the year prior to benralizumab, and higher initial blood eosinophil levels. A significant association was also found with the patient’s sex, specifically with women (the values are shown in detail in Appendix A). After the multivariate analysis was performed, a significant association was found with fewer OCS bursts during the year prior to biological therapy (OR = 0.74; 95% CI = 0.52–0.93), higher blood eosinophil levels (OR = 1.004; 95% CI = 1–1.01), and women (OR = 5.20; 95% CI = 1.34–23.58). The results of the multivariate analysis are shown in detail in Table 7.

##### Response to Reduction in Oral Corticosteroid Maintenance (OCS)

In the bivariate analysis, an association was found between successful response and lower OCS bursts in the previous year, lower baseline FEV1 values, and no maintenance OCS (values can be found in detail in Appendix A). In the multivariate analysis, lower baseline FEV1 values (OR = 0.94; 95% CI = 0.87–0.99; Table 7) and fewer OCS bursts (OR = 0.71; 95% CI = 0.50–0.92; Table 7) were associated as predictors of reduced maintenance OCS.

##### Response to Lung Function (FEV1)

In the bivariate and multivariate analysis, an association was found between improvement of lung function after 12 months of treatment with benralizumab and lower initial FEV1 values (OR = 0.96; 95% CI = 0.92–0.98; Table 7). The bivariate analysis values are shown in detail in Appendix A.

##### Response to Reduction of Exacerbations

The bivariate analysis showed a statistically significant association between the reduction of exacerbations and patients who were non-smokers (the values can be found in detail in Appendix A). In the multivariate analysis, the association remained significant (OR = 2.89; 95% CI = 1.89–4.30; Table 7).

## 3. Discussion

Several RCTs have demonstrated that mepolizumab and benralizumab are safe and effective in patients with severe uncontrolled asthma [35,38,39,40,41]. However, real-life data may differ from those obtained from RCTs because of the emphasis on internal validity due to standardization and control protocols that may compromise external validity, and therefore, extrapolation of results. Real-life studies provide complementary information to RCTs, which supports clinical decision driving to a better comprehension of the effectiveness of the drug in real clinical practice conditions and to achieve a more personalized approach. 

In this study, we found that mepolizumab treatment in severe uncontrolled asthma was very effective, significantly reducing OCS bursts per year, maintenance OCS, blood eosinophil levels, IgE, and annual rate of exacerbations requiring emergency department treatment and/or hospitalization. The reduction in IgE levels was significant, however, as expected, it was not a biomarker of response for this anti-IL5 therapy. A significant improvement in asthma control occurred, with increases in ACT questionnaire scores and FEV1. The MENSA and MUSCA clinical trials showed a clinically significant reduction in exacerbation rates by 53% and 58% (*p* < 0.001), respectively [38,39]. In our study, annual bursts of OCS are a proxy for the overall exacerbation rate and were significantly reduced by 38.5%. The rate of exacerbations requiring emergency or hospitalization in the MENSA study were reduced by 61% and FEV1 was significantly improved (*p* = 0.03) [38]. The results of our study resemble these data, with a 74% reduction in exacerbations requiring emergency or hospitalization and a significant improvement in mean FEV1 of 10.3%. Real-life studies also reported improvements in lung function, asthma control, reduced the usage of OCS bursts, and maintenance OCS, and decreased eosinophil counts and exacerbations [29,31,42,43,44,45,46,47]. Among these studies, we found a systematic review by Israel E. et al. (2021) reflecting similar results in both prospective and retrospective studies [42]. Consistent with our results, the retrospective studies included in the review showed significant reductions of 54% to 94% (*p* < 0.001 and *p* = 0.0012, respectively) in OCS bursts; reductions of 55% (*p* < 0.001) in exacerbations requiring emergency or hospitalization. The 73% (*p* = 0.004) and 83% (*p* values not reported) showed reductions in the rate of exacerbations requiring hospitalization; 27% to 84% had discontinued maintenance OCS and from 32% to 100% had reduced the initial maintenance dose (*p* < 0.001); lung function improved by 3% to 8% (*p* < 0.05); reductions in blood eosinophilic counts from 69% to 92% (*p* < 0.05); and changes in ACT from 5 to 8 points [42]. However, there are studies in which an improvement in lung function was not achieved, which may be due to the difference in population, history of smoking, or smaller sample size [45,46,47]. 

The markers significantly associated with the corticosteroid-saving effect were the absence of exacerbations and lower ICS levels; the improvement in lung function was related to low initial FEV1 percentages and absence of SAHS, and the absence of exacerbations during the year prior to treatment was associated with a positive response to the reduction in exacerbations. Previous studies relate high eosinophil levels to a greater treatment response with mepolizumab [48,49]; these studies are not comparable with our results since they evaluate the overall response to treatment.

After 12 months of treatment with benralizumab, very promising results were reported, with substantial reductions in OCS bursts, blood eosinophils, and number of exacerbations requiring emergency care or hospitalization. Moreover, lung function and ACT score experienced significant improvements. The results presented are in accordance with the RCTs: ZONDA, SIROCO, and CALIMA. In our study, a significant reduction of exacerbations requiring emergency or hospitalization of 77% was observed, while the SIROCCO and CALIMA studies reflect reductions of 42% and 36% (*p* < 0.001), respectively. Clearly, the reduction is greater in our study, which may be due to the greater severity of the patients included in the randomized trials. Our results reflect a 77% reduction in the rate of exacerbations, similar to the ZONDA study where reductions of 70% (*p* < 0.001) are reflected. Real-life studies in recent years are in line with our investigation [50,51,52,53]. Among these studies, a recent systematic review and meta-analysis conducted by Charles, D. et al., 2022, shows that benralizumab produces significant changes in the annual rate of exacerbations and FEV1 percentage, significant improvement in asthma symptoms measured by ACT, and a decrease in blood eosinophilia [32].

Benralizumab was associated with a greater OCS saving in women, patients with higher baseline eosinophil levels, and fewer OCS cycles during the previous year. As in the case of previous biological therapies, the improvement in lung function was associated with lower initial FEV1 levels, and greater reduction in exacerbations with never-smoking patient profiles. The main real-life study that reports response markers relates the absence of exacerbations and OCS to a strongly eosinophilic phenotype and lesser severity of the disease [54]. These markers are consistent with those obtained in our study for the absence or reduction of OCS after 12 months of biological therapy with benralizumab. However, the markers do not resemble those obtained in our study for reduction of exacerbations and improvement in lung function. This may be because these authors analyzed total treatment response and did not distinguish between OCS reduction, exacerbations reductions, and lung function improvement, as in our study.

It is worth emphasizing that we did not find statistically significant differences after 12 months of treatment in the group of patients with FEV1 greater than 80% for any of the two biological therapies, which suggests that they improve lung function, but that in most cases, they do not reach values higher than 80% of FEV1 after the first year of treatment.

The main limitation of this study, like all real-life investigations, is the lack of a placebo control group. Its absence means that the magnitude of the results observed lacks the firmness of comparison with a control group. In addition, it has limitations inherent to a retrospective data collection, such as the lack of relevant values for some patients such as ACT and baseline spirometry, the sample size, and the impossibility of extending the study with other anti-IL5 therapies, such as reslizumab, due to the lack of patients in the hospital with this treatment. However, real-life studies contribute extensive knowledge of biological therapies and enable the findings to be implemented in clinical practice.

## 4. Materials and Methods

We conducted a real-life retrospective observational cohort study.

### 4.1. Study Population

This study included 172 patients over the age of 18 years of Caucasian origin diagnosed with SUA according to the criteria of the Spanish Asthma Management Guidelines (GEMA 5.1), recruited in the Respiratory Medicine Department of the Hospital Universitario Virgen de las Nieves in Granada (Spain) between April 2014 and April 2022. Of the 172 patients, 103 were candidates for mepolizumab and 69 for benralizumab (Figure 5). Of the 103 mepolizumab candidates for inclusion in the study, the effectiveness of mepolizumab was finally evaluated in 89 (86.4%) patients. A total of 8.7% (9/103) of the patients did not complete the 12 months of treatment with mepolizumab; in 3.9% (4/103), one of the essential clinical variables was not evaluated; and 1% (1/103) had an adverse event. The response was evaluated in 57 (82.6%) patients out of the 69 candidates for participation in the benralizumab study. Of the latter, 15.9% (11/69) had not completed 12 months of treatment when the data were collected and 1.4% (1/69) had no evaluation for one of the essential variables for the study. The administration route of the drug was subcutaneous: 100 mg de mepolizumab every 4 weeks and 30 mg of benralizumab every 4 weeks for the first three doses and subsequently every 8 weeks [55,56]. The remaining patients did not meet the study’s evaluation criteria.

### 4.2. Socio-Demographic and Clinical Variables

The socio-demographic variables included age, sex, body mass index (BMI), smoking status, years with the disease, nasal polyps, previous respiratory disease, allergies, gastroesophageal reflux disease (GERD), sleep apnea-hypopnea syndrome (SAHS), chronic obstructive pulmonary disease (COPD), years with the treatment, treatment dose, and change to another monoclonal antibody therapy. Individuals were classified as non-smokers if they had never smoked or had smoked fewer than 100 cigarettes in their lives, as former smokers if they had smoked 100 or more cigarettes in their lives but did not currently smoke, and as current smokers. For BMI, following the WHO criteria, individuals were classified as underweight (BMI < 18.5), healthy weight (18.5 < BMI < 24.9), overweight (25 < BMI < 29.9), or obese (BMI > 30) [57].

The clinical variables were collected according to the year before starting treatment with the biological therapy and after completing the first year of treatment. They included ICS maintenance doses expressed as fluticasone propionate µg equivalents, OCS bursts required in the follow-up period and maintenance OCS doses expressed as mg prednisone equivalents [58], blood eosinophil count, exacerbations requiring emergency department treatment and/or hospitalization with OCS for at least 3 days, IgE, lung function as percentage forced expiratory volume in the first second (%FEV1), and asthma control test (ACT) [59].

### 4.3. Statistical Analysis

The quantitative variables were expressed as mean (± standard deviation) for those that complied with normality and as median and percentiles (25 and 75) for those that did not follow a normal distribution. Normality was confirmed using the Kolmogorov-Smirnov test.

The clinical variables responsible for the response were compared before and after treatment using the McNemar test for qualitative variables. For quantitative variables that complied with normality, we used the *t* test for paired data and the Mann-Whitney U test (Wilcoxon rank sum test) for non-normal variables. The results were considered significant when the *p* value was less than 0.05.

To evaluate the predictors of response at 12 months, the following were taken as response variables: reduction of bursts of OCS per year, considering a reduction of at least 50% in the bursts or absence of OCS bursts as a satisfactory response; reduction of maintenance OCS, considering a reduction of at least 50% in the maintenance OCS or absence as a satisfactory response; improvement of lung function, considering those that achieved an increase of at least 10% in FEV1 after 12 months’ treatment as responsive; and reduction of exacerbations per year requiring emergency department treatment and/or hospitalization, taking a reduction of at least 50% or absence of exacerbations as a satisfactory response. The bivariate analysis between the response and socio-demographic and clinical variables was performed using Pearson’s chi-squared test or applying Fisher’s exact test for the qualitative variables. For the quantitative variables, Student’s *t*-test was applied to the variables that complied with normality. The Mann–Whitney U test was applied for non-normal variables. A multivariate (logistic regression) analysis was used to calculate the adjusted odds ratio (OR) and the 95% confidence interval (CI) for the possible prognostic factors of response to OCS bursts, maintenance OCS, lung function, and exacerbations. All the variables that were significant in the bivariate analysis were included in the model for the multivariate analysis. In the multivariate analysis, all variables that were not significant were eliminated, thus obtaining the final model for each type of response. All the tests were two-sided, with a probability of 0.05 or less considered statistically significant, and were performed with the R 4.2.0 free software.

## 5. Conclusions

The efficacy and safety of biological therapies have been extensively studied and demonstrated in controlled clinical trials, but the results reported by this real-life study are very promising. We show how mepolizumab and benralizumab significantly improve lung function and reduce and/or prevent the presence of exacerbations and the use of oral corticosteroids in a large proportion of patients. Moreover, the search for predictive factors of response to biological therapies in real life may provide information for decisions on their clinical management. We can therefore conclude that the biological therapies studied have had a great impact on the quality of life of patients with severe uncontrolled asthma and the associated healthcare burden.

## Figures and Tables

**Figure 1 ijms-24-02011-f001:**
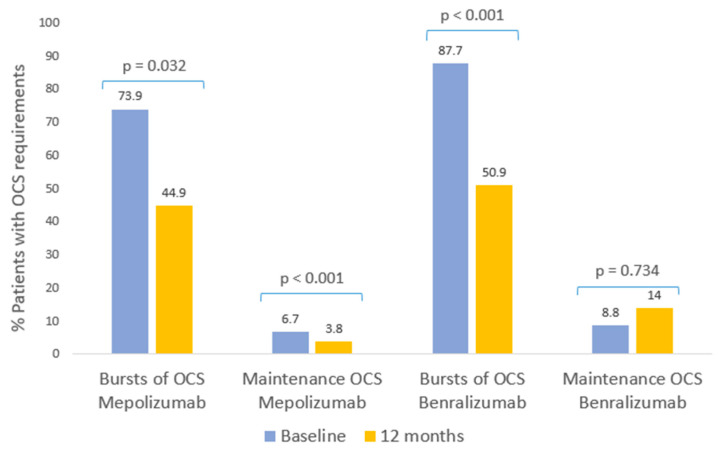
Clinical response to biological therapies: bursts of OCS and maintenance OCS. Patients with bursts of oral corticosteroid requirements and maintenance OCS during the 12 months prior to starting mepolizumab and benralizumab, and at the follow-up evaluation after 12 months of therapy.

**Figure 2 ijms-24-02011-f002:**
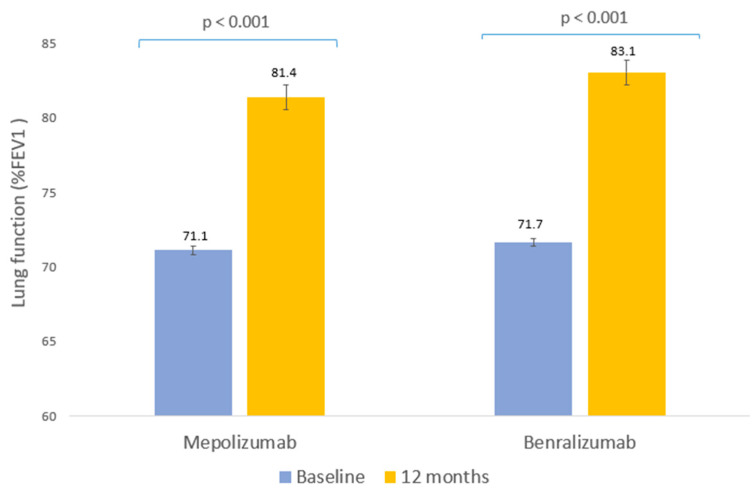
Clinical response to biological therapies: lung function. Mean percentage of peak forced expiratory volume in the first second of forced exhalation (FEV1) prior to biological therapy and after 12 months of therapy.

**Figure 3 ijms-24-02011-f003:**
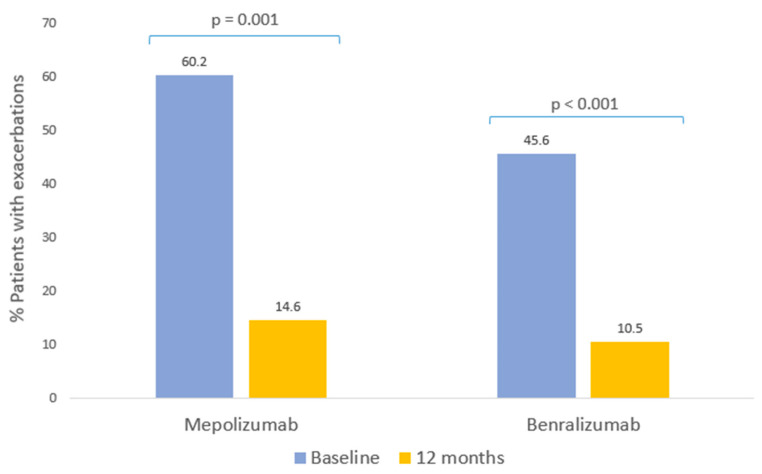
Clinical response to biological therapies: exacerbations. Patients with at least one exacerbation in the 12 months prior to starting mepolizumab and benralizumab, and at follow-up evaluation after 12 months of therapy.

**Figure 4 ijms-24-02011-f004:**
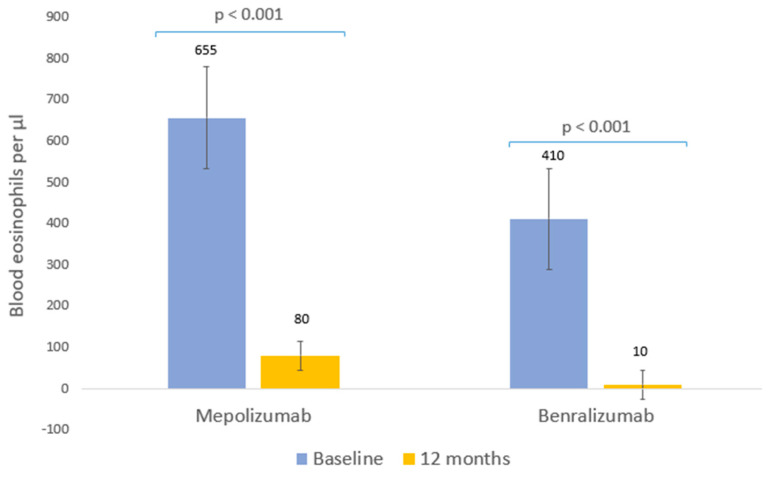
Clinical response to biological therapies: blood eosinophils. Median eosinophils prior to biological therapy and after 12 months of treatment.

**Figure 5 ijms-24-02011-f005:**
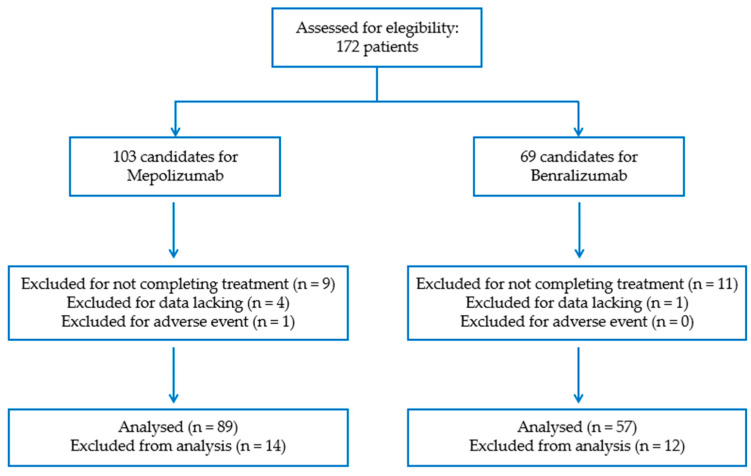
Patient eligibility criteria.

**Table 1 ijms-24-02011-t001:** Demographic and clinical characteristics of patients treated with mepolizumab.

	N	n (%)
Age	89	55.8 ± 13.1
Sex	89	
Women		58 (65.2)
Men		31 (34.8)
BMI	89	
Underweight		4 (4.5)
Normal weight		19 (21.3)
Overweight		38 (42.7)
Obese		28 (31.5)
Tobacco consumption	89	
Non-smoker		74 (83.2)
Former smoker		14 (15.7)
Current smoker		1 (1.1)
Previous respiratory disease	89	
Yes		38 (42.7)
No		51 (57.3)
Polyps	89	
Yes		38 (42.7)
No		51 (57.3)
Allergies	89	
Yes		44 (49.4)
No		45 (50.6)
GERD	89	
Yes		35 (39.3)
No		54 (60.7)
SAHS	89	
Yes		16 (18)
No		73 (82)
COPD	89	
Yes		14 (15.7)
No		75 (84.3)
Years with EA	89	5 [3–10]
ICS (µg/day)	89	500 [500–1000]
Bursts of OCS per year	89	2 [0–4]
Yes		65 (73)
No		24 (27)
Bursts of OCS (Yes)	65	3 [2–5]
Maintenance OCS (mg/day)	89	0 [0–0]
Yes		6 (6.7)
No		83 (93.3)
Maintenance OCS (Yes)	6	11.3 [5.6–15]
Baseline %FEV1	85	71.1 ± 23.7
<80		60 (70.6)
>80		25 (29.4)
Baseline ACT	28	12.5 [9–15.8]
Exacerbation in previous year	89	1 [0–2]
Yes		53 (60.2)
No		35 (39.8)
Exacerbation in previous year (Yes)	53	2 [1–3]
Baseline blood eosinophil count (cells/μL)	89	655 [330–905]
Baseline IgE (IU/mL)	54	112.6 [30.3–279.8]
Years with mepolizumab	89	2 [1–4]
Previous BT	89	
Yes		23 (25.8)
No		66 (74.2)
Change of BT	89	
Yes		18 (22.2)
No		71 (79.8)

BMI: body mass index; GERD: gastroesophageal reflux disease; SAHS: sleep apnea-hypopnea syndrome; COPD: chronic obstructive pulmonary disease; EA: eosinophilic asthma; ICS: inhaled corticosteroids; OCS: oral corticosteroids; FEV1: maximum expiratory volume in the first second of forced expiration; ACT: asthma control test; IgE: immunoglobulin E; BT: biological therapy. ICS dose is expressed as µg/day fluticasone equivalents. Qualitative variables are shown as numbers (percentage, %). Quantitative variables with normal distribution are shown as mean ± standard deviation (SD). Quantitative variables with non-normal distribution are shown as p50 (p25–p75).

**Table 2 ijms-24-02011-t002:** Demographic and clinical characteristics of patients treated with benralizumab.

	N	n (%)
Age	57	58.4 ± 14.3
Sex	57	
Women		37 (64.9)
Men		20 (35.1)
BMI	57	
Underweight		0 (0)
Normal weight		12 (21.1)
Overweight		19 (33.3)
Obese		26 (45.6)
Tobacco consumption	57	
Non-smoker		42 (73.7)
Former smoker		14 (24.6)
Current smoker		1 (1.7)
Previous respiratory disease	57	
Yes		26 (45.6)
No		31 (54.4)
Polyps	57	
Yes		22 (38.6)
No		35 (61.4)
Allergies	57	
Yes		35 (61.4)
No		22 (38.6)
GERD	57	
Yes		23 (40.4)
No		34 (59.6)
SAHS	57	
Yes		10 (17.5)
No		47 (82.5)
COPD	57	
Yes		11 (19.3)
No		46 (80.7)
Years with EA	57	7 [4–9]
ICS (µg/day)	57	1000 [500–1000]
Bursts of OCS per year	57	2 [1–4]
Yes		50 (87.7)
No		7 (12.3)
Bursts of OCS per year (Yes)	50	3 [1.3–4]
Maintenance OCS (mg/day)	57	0 [0–0]
Yes		5 (8.8)
No		52 (91.2)
Maintenance OCS (Yes)	5	10 [10–15]
Baseline %FEV1	57	71.7 ± 22.5
<80		38 (66.7)
>80		19 (33.3)
Baseline ACT	27	13 [9.5–15.5]
Exacerbation in previous year	57	0 [0–1]
Yes		26 (45.6)
No		31 (54.4)
Exacerbation in previous year (Yes)	26	1 [1–2]
Baseline blood eosinophil count (cells/μL)	57	410 [230–570]
Baseline IgE (IU/mL)	45	137.7 [50–758]
Years with mepolizumab	57	2 [1–3]
Previous BT	57	
Yes		19 (33.3)
No		38 (66.7)
Change of BT	57	
Yes		5 (8.8)
No		52 (91.2)

BMI: body mass index; GERD: gastroesophageal reflux disease; SAHS: sleep apnea-hypopnea syndrome; COPD: chronic obstructive pulmonary disease; EA: eosinophilic asthma; ICS: inhaled corticosteroids; OCS: oral corticosteroids; FEV1: maximum expiratory volume in the first second of forced expiration; ACT: asthma control test; IgE: immunoglobulin E; BT: biological therapy. ICS dose is expressed as µg/day fluticasone equivalents. Qualitative variables are shown as numbers (percentage, %). Quantitative variables with normal distribution are shown as mean ± standard deviation (SD). Quantitative variables with non-normal distribution are shown as p50 (p25–p75).

**Table 3 ijms-24-02011-t003:** Changes in baseline variables after 12 months of biological therapy.

Independent Variable	Biological Therapy	
	Mepolizumab	Benralizumab
ICS dose (µg/day)		
Baseline, p_50_(p_25_, p_75_)	500 [500–1000]	1000 [500–1000]
Follow-up, p_50_(p_25_, p_75_)	500 [250–1000]	500 [500–1000]
Change from baseline	*p* = 0.218	*p* = 0.039
Bursts of OCS (Yes/No)		
Baseline, n (%)	65 (73.9)	50 (87.7)
Follow-up, n (%)	40 (44.9)	23 (50.9)
Change from baseline	*p* = 0.032	*p* < 0.001
Bursts of OCS per year		
Baseline, p_50_(p_25_, p_75_)	2 [0–4]	2 [1–4]
Follow-up, p_50_(p_25_, p_75_)	0 [0–2]	1 [0–2]
Change from baseline	*p* < 0.001	*p* = 0.001
Maintenance OCS (Yes/No)		
Baseline, n (%)	6 (6.7)	5 (8.8)
Follow-up, n (%)	3 (3.8)	8 (14)
Change from baseline	*p* < 0.001	*p* = 0.377
Maintenance dose of OCS per year		
Baseline, p_50_(p_25_, p_75_)	0 [0-0]	0 [0-0]
Follow-up, p_50_(p_25_, p_75_)	0 [0-0]	0 [0-0]
Change from baseline	*p* = 0.095	*p* = 0.734
Patients with FEV1 > 80%		
Baseline, n (%)	17 (28.3)	19 (33.3)
Follow-up, n (%)	20 (40)	25 (50)
Change from baseline	*p* = 0.058	*p* = 0.366
FEV1 (%)		
Baseline, mean (SD)	71.1 ± 23.7	71.7 ± 22.3
Follow-up (SD)	81.4 ± 18	83 ± 24
Change from baseline	*p* < 0.001	*p* < 0.001
ACT		
Baseline, p_50_(p_25_, p_75_)	12.5 [9–15.8]	13 [9.5–15.5]
Follow-up, p_50_(p_25_, p_75_)	22 [19–23]	22 [19–24]
Change from baseline	*p* < 0.001	*p* < 0.001
Presence of exacerbations (Yes/No)		
Baseline, n (%)	53 (60.2)	26 (45.6)
Follow-up, n (%)	13 (14.6)	6 (10.5)
Change from baseline	*p* = 0.001	*p* < 0.001
Exacerbations per year		
Baseline, p_50_(p_25_, p_75_)	1 [0–2]	0 [0–1]
Follow-up, p_50_(p_25_, p_75_)	0 [0–0]	0 [0–0]
Change from baseline	*p* < 0.001	*p* < 0.001
Blood eosinophils (cells/μL)		
Baseline, p_50_(p_25_, p_75_)	655 [330–905]	410 [230–570]
Follow-up, p_50_(p_25_, p_75_)	80 [40–120]	10 [10–10]
Change from baseline	*p* < 0.001	*p* < 0.001
IgE (IU/mL)		
Baseline, p_50_(p_25_, p_75_)	112.6 [30.3–279.8]	137.7 [50–758]
Follow-up, p_50_(p_25_, p_75_)	79 [34.9–204.1]	136 [51–147]
Change from baseline	*p* = 0.015	*p* = 0.385

ICS: inhaled corticosteroids; OCS: oral corticosteroids; FEV1: maximum expiratory volume in the first second of forced expiration; ACT: asthma control test; IgE: immunoglobulin E. ICS dose is expressed as µg/day fluticasone equivalents. Qualitative variables are shown as numbers (percentage, %). Quantitative variables with normal distribution are shown as mean ± standard deviation (SD). Quantitative variables with non-normal distribution are shown as p_50_ (p_25_–p_75_).

**Table 4 ijms-24-02011-t004:** Clinical effectiveness of mepolizumab in patients with severe uncontrolled asthma.

Response Definition	n	%
Bursts OCS reduction ≥ 50%		
Yes	40	44.9
No	29	32.6
No OCS	20	22.5
Bursts OCS reduction ≥ 50% or absence		
Yes	60	67.4
No	29	32.6
Maintenance OCS reduction ≥ 50%		
Yes	4	4.5
No	2	2.2
No maintenance OCS	83	93.3
Maintenance OCS reduction ≥ 50% or absence		
Yes	87	97.8
No	2	2.2
FEV1 increase ≥ 10%		
Yes	38	48.1
No	41	51.9
FEV1 increase ≥ 10% or FEV1 ≥ 80%		
Yes	55	67.9
No	26	32.1
Exacerbation reduction ≥ 50%		
Yes	42	47.2
No	10	11.2
No exacerbations	37	41.6
Exacerbation reduction ≥ 50% or absence		
Yes	79	88.8
No	10	11.2

OCS: oral corticosteroids; FEV1: maximum expired volume in the first second of forced expiration.

**Table 5 ijms-24-02011-t005:** Clinical effectiveness of benralizumab in patients with severe uncontrolled asthma.

Response Definition	n	%
Bursts OCS reduction ≥ 50%		
Yes	29	50.9
No	22	38.6
No OCS	6	10.5
Bursts OCS reduction ≥ 50% or absence		
Yes	35	61.4
No	22	38.6
Maintenance OCS reduction ≥ 50%		
Yes	3	5.3
No	5	8.8
No maintenance OCS	49	85.9
Maintenance OCS reduction ≥ 50% or absence		
Yes	52	91.2
No	5	8.8
FEV1 increase ≥ 10%		
Yes	25	52.1
No	23	47.9
FEV1 increase ≥ 10% or FEV1 ≥ 80%		
Yes	35	71.4
No	14	28.6
Exacerbation reduction ≥ 50%		
Yes	24	42.1
No	3	5.3
No exacerbations	30	52.6
Exacerbation reduction ≥ 50% or absence		
Yes	54	94.7
No	3	5.3

OCS: oral corticosteroids; FEV1: maximum expired volume in the first second of forced expiration.

**Table 6 ijms-24-02011-t006:** Predictors of response after 12 months of treatment with mepolizumab in patients with severe uncontrolled asthma (multivariate analysis).

	B	Odds Ratio	*p*-Value	95% CI
OCS bursts reduction predictors				
ICS dose	−0.0015	0.99	0.037	0.99–0.99
Exacerbation in previous year	−0.6993	0.47	<0.001	0.33–0.70
Maintenance OCS reduction predictors				
Maintenance OCS (mg/day)	−0.3095	0.73	0.008	0.55–0.91
Lung function improvement predictors				
SAHS (No)	2.6701	14.44	0.013	2.05–146.15
FEV1	−0.1127	0.89	<0.001	0.84–0.93
Exacerbation reduction predictors				
Absence of exacerbations in the previous year	−1.939	0.14	0.041	0.01–0.82

OCS: oral corticosteroids; ICS: inhaled corticosteroids; SAHS: sleep apnea-hypopnea syndrome; FEV1: maximum expired volume in the first second of forced expiration; 95% CI: 95% confidence interval.

**Table 7 ijms-24-02011-t007:** Predictors of response after 12 months of treatment with benralizumab in patients with severe uncontrolled asthma (multivariate analysis).

	B	Odds Ratio	*p*-Value	95% CI
OCS bursts reduction predictors				
Bursts of OCS per year	−0.3049	0.74	0.028	0.52–0.93
Eosinophils	0.0036	1.01	0.019	1.00–1.01
Sex (Female)	1.6486	5.20	0.002	1.34–23.58
Maintenance OCS reduction predictors				
Bursts of OCS per year	−0.3436	0.71	0.016	0.50–0.92
FEV1	−0.0649	0.94	0.05	0.87–0.99
Lung function improvement predictors				
FEV1	−0.0453	0.96	0.007	0.92–0.98
Exacerbation reduction predictors				
Tobacco consumption (Non-smoker)	−1.939	2.89	0.011	1.89–4.30

OCS: oral corticosteroids; FEV1: maximum expired volume in the first second of forced expiration; 95% CI: 95% confidence interval.

## Data Availability

Not applicable.

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
