# Peer review of "Impact of Anti-IL5 Therapies on Patients with Severe Uncontrolled Asthma and Possible Predictive Biomarkers of Response: A Real-Life Study"

_ijms, 2023, doi:10.3390/ijms24032011_

Round 1

Reviewer 1 Report

General comments:

Dear authors, I have read your manuscript  »Impact of Anti-IL5 Therapies on Patients with Severe Uncontrolled Asthma and Possible Predictive Biomarkers of Response: A Real-Life Study« carefuly.

It is a huge work with many data and promising title. It is true that a real life data are missing.

However I put comments and recommendations especialy from the clinical point of view with the aim of improving  the text and better understanding of these important data and response to biologics.

Specific comments:

1.     Please explain SAHS and COPD in the text (line 94), as they are mentioned for the first time.

2.     Line 95, 96: »All the patients had required ICS during the previous year, with a median of 225 [184–640] mg/day, and 73.86% (65/89) needed OCS, with a median of 2 [0–4] cycles.«

In my opinion it might be not clear enough to the reader.

Concerning ICS:  Is the dosing of ICS (inhaled corticosteroids) and especialy its unit (miligram/day) correct?  Which is the inhaled corticosteroid or its equvivalent; fluticasone, beclomethasone…? Please, be so kind and check the dosing and the type of inhaled corticosteroid (all inhaled steroids have to be calculated to one type of inhaled corticosteroid).

Concering OCS (….and 73.86% (65/89) needed OCS): It is not clear wheather you are talking about mainenance OCS or bursts of OCS due to exacerbation. Please, it is very important to separate these two options; bursts are indeed linked to exacerbations. In maintenance OCS is important the basic and follow up dosing-important cornerstone in efficacy of biologics (evaluation of OCS load lowering).

3.     Line 100, 101: »A total of 25.84% (23/89) had previously received another monoclonal antibody therapy and 22.22% (18/89) required changing to  another biological therapy for asthma control«

Please, it is not clear: what does it mean the statement that 22,22% required changing to  another biological therapy for asthma control. You changed mepo to another biologic in nonresponders?

4.     Table 1: please explain how underweight, normal, overweight and obese are defined according to BMI (in the table).

Since you have a table, maybe you could explain the results shortly in the text.

Table 3 and 4:

-Why you have decided to use 10% FEV1 change to be important.

OCS reduction ≥ 50% or absence«- do you mean withdrawal of OCS (with »absence«)?

An important add on would be to make clear the difference among OCS maintenence (with dosing) and OCS bursts at this point.

Fig 1: OCS treatment: please be so kind and explain is this OCS maintenance? If yes, it might be not clear enough for the reader.

5.

2.3.7. Immunoglobulin E

Is this point clinicaly relevant in anti IL 5 therapy? Please comment in the discussion.

6. Line 227:

2.4.1.1. Response to reduction of oral corticosteroids (OCS) . The bivariate analysis showed a satisfactory response associated with lower initial  ICS values, absence of OCS cycles during the year prior to mepolizumab..«

The definition of severe asthma include exacerbations with OCS bursts despite high dosing of ICS. What was the reason for zero exacerbations? Were these patients on maintenence OCS and consequently controlled with OCS maintenance?

7. Please correct the discussion according to previous recommended changes.

Thank you

Author Response

Responses to comments can be found in the attached file.

Reviewer 2 Report

This study examines the "real world" effects of meprolizumab and benralizumab in a single-center referral hospital.  These are recently approved medicines and so this is a clinically relevant study.  However, this study has multiple limitations that the authors should address to strengthen this manuscript.  These are outlined below.

Major Comments

1. The rationale for the study is not well-described.  Although the authors state that post-marketing studies for these medications have been "limited", there have been several studies examining the effects of meprolizumab and benralizumab in asthmatics since the original trials.  The authors need to be more clear regarding what the prior "real world" studies did not elucidate about these two medications and what gap their study was designed to fill.

2.  The Results are very confusing.  For example, the data regarding oral corticosteroid use in meprolizumab patients states that 65 of 89 patients needed oral corticosteroids.  Thus, 24 patients did not require oral corticosteroids (page 6).  After a year of meprolizumab, this number had decreased to 20 patients (page 6).  However, Table 5 lists 40 patients as not having required oral corticosteroids.  It is not possible to determine if the authors made a mistake in the report of corticosteroid use or if the data are being presented in such a way as to make it exceptionally difficult to read correctly.  This is only one example of the confusing data presentation in the Results section - there are several additional instances of the data being very challenging to read for both the meprolizumab and benralizumab patients.  The Results need to be revised to be clear and easy for the interested reader to follow.

3. The Discussion only briefly mentions other studies examining the post-marketing experience with meprolizumab and benbralizumab.  These studies need to be reviewed in greater detail.  Specifically, the authors need to provide more information as to how their study differs from prior work so that the interested reader can obtain a full picture regarding how this study adds to the knowledge about asthma therapies.

Minor

1. All of the abbreviations used in the manuscript should be written in the text as the non-abbreviated term with the abbreviation indicated in parenthesis and then the abbreviation used afterwards.  This needs to be corrected for both GERD and SAHS.

2. The supplemental tables list the terms "unsatisfactory" and "successful" but do not provide the definitions for these terms in the legends.  The definitions should be included in the legends.

Author Response

(The authors gave the same response as above.)

Reviewer 3 Report

The paper "Impact of Anti-IL5 Therapies on Patients with Severe Uncontrolled Asthma and Possible Predictive Biomarkers of Response: A Real-Life Study", by Rojo-Tolosa et al, describes real-life data on the use of two anti-IL5 for asthma treatment. The paper describes interesting findings, that support previous data from clinical trials. Neverthless,  there are some aspects that should be revised.

Major comments:

- Introduction, line 81 - the study aims are not described in the manuscript.

- Methods section: it must be improved to include information regarding eligibility criteria.  Definitions should be clearer, especially the definition used for exacerbations - only worsening with need for ER and/or hospitalizations were included? Even if they had no need for OCS treatment? And if an exacerbation required corticosteroids but no emergency department, is it included here? Please include a reference to support your definition. A description regarding how you merged the dose data from different ICS and OCS should be presented (including at least a reference to the equivalence table that was used and the reference ICS and OCS used in the tex).

- Methods, statistical analysis: Please check this section. A multivariable linear regression does not give OR (and considering the description you provide, the dependent variables were binary which would make linear regression inapropriate). Moreover, this analysis must be more clearly described, namely, how did you select the variables to include in the model? How did you adjust the model until the final one? Which tests were used to (at least) assess model accuracy and goodness of fit?

- Discussion section should be improved. Other limitations should be stated. It is a retrospective study and has all the limitations that are inherent to a retrospective data collection (including missing data in relevant variables like ACT). Moreover, only 2 anti-IL5 were included - please discuss why reslizumab was not included. 

Minor comments:

- Abstract, line 22 - SEA should be SUA

- introduction, line 54 - it is not clear to which data "These data" reports. Consider specifying.

- introduction, line 60-62: Consider improving this sentence by including a definition of asthma exacerbations considering current guidelines.

- Introduction, lines 66-73 - consider splitting the sentence to improve readability

- Introduction, line 76 - it is not clear to which this "it" refers to. Consider clarifying.

- introduction, line 80 - maybe "In contrast" is not the best expression (it seems that the results are conflicting

- results and tables: Consider reducing the number of decimals (leaving 1 at most) in percentages, means (SD). Use the same number of decimals in all data of the same type. 

- results (e.g. lines 85-101, lines 112-128): in the results section avoid repeating the data presented in the table. Describe in the text only those data that are most relevant to this study aims (and the rest can be seen in the table). That will improve readability.

- Results and methods (e.g. lines 95-96): ICS are not all the same. It should be clear which ICS you are using to present the doses. Please check my major comment regarding this issue.

- results, line 100: were they all uncontrolled? If not, consider including ACT classification as a categorical variable (controlled or uncontrolled), both in the baseline assessment and follow-up.

 - results, lines 100-102 and 126-128: the same information is described in lines 88-90 and 115-116, respectively. Please remove or change or clarify the need to repeat.

- Tables 1 and 2: as you have many blank cells in the column in the right, consider merging this column with the N and %, indicating in the table title that "results are presented as n% except when otherwise indicated" and stating in the variables presented as mean or median how they are being presented. You may include data regarding missings as footnote and the total n in the table title. 

- Table 1, OCS cycles: the median[P25-P75] is for the total number of patients, including those that did not use OCS? Consider having this information as a "subsection" of those who used at least once ("Yes") instead of the total. Consider the same for asthma exacerbations.

- Results, clinical effectiveness subsection: I would suggest having this information after 2.3.  Comparison of clinical variables before and after treatment. This section benefits from some of the information presented there and will be easier to understand with those data.

- Results, lines 140-144 and 154-157: if kept, this information should be presented in the beginning of the methods. At this point the authors have already described the 89 mepo / 57 benra patients included and there is no point in stating that there were 103 / 69 candidates for inclusion. Nevertheless, I would suggest presenting a patient inclusion flowchart at the beginning of the methods (including those under mepolizumab and those under benralizumab, and the reasons for non-inclusion). The eligibility criteria should be described in the methods.

- Table 3: consider OCS withdrawn instead of absence. Several patients had FEV1 >=80% at baseline; how were they considered in the "FEV1 increase >=10% or FEV1 >=80%"?

- Results, line 164-165: in the sentence "52.63% suffered no exacerbation requiring emergency department treatment and/or hospitalization" It is not clear if in the previous uses the meaning was the same; please clarify / include the definition of exacerbations in the methods.

- Figure 1 and 3 - include the unit in the yy axis. Consider using the percentage instead of the n. 

- Figure 2 and 4 - I suggest that the authors include the error bars in the graphs representing continuous data. 

- Figure 2 - consider presenting the data stratified according to the baseline FEV1 (<80%  vs >=80%)

- Figure 3 - The legend should be more clear in stating that you are representing patients with at least one asthma exacerbation.

- Results, lines 222-224 - As this data is already presented in the table I would remove this subheading

- Results, subheading "predictors of response": consider including the criteria for response in the subtitle or text to make it easy to follow without the need to go back. This might be unnecessary if you change the order of the subheadings putting the data regarding clinical effectiveness closer to this section. 

- Results, line 233: Consider changing " indicated a tendency towards a better therapeutic response" to "were significantly associated with a better therapeutic response"

- Table 6: consider changing ICS to "ICS dose", exacerbation to "absence of  exacerbations in the previous year" (check if this is the correct interpretation as this section is not clear in reporting if exacerbation data was included as number or presence / absence). Consider changing from "lung improvement predictors" to "lung function improvement predictors". This table should be more clear regarding the variables that were included in the model and how they can be interpreted. The table should be interpretable on its own - check my two following comments that might help clarify some of my doubts.

- results, lines 244-245: With the information in table 6 I would interpret that the presence of SAHS is associated with a higher odds of improvement. Please check and change the table so that the presented data can be interpreted adequately. SAHS should be included in the model as a categorical variable and the table must state which category is being use to state the OR.

- results, lines 150-152 - the text and table 6 should be more clear regarding the variable that was included in the model (number of exacerbations or presence / absence of exacerbations) - please clarify the text and make the data in the table more understandable. 

- Table 7: it has the same problems reported regarding table 6. Please clarify these variables / categories so that the table can be understood on its own (it is not possible to know which sex is associated with increased risk or if OCS was used as number or presence / absence or which category is represented by smoking)

- all sections: make the use of the acronym SUA consistent throughout the text

- Improve English throughout the text. 

Author Response

(The authors gave the same response as above.)

Reviewer 4 Report

OCS median dose is not available. In figure 1 we are seeing only the patients who are or are not on OCS but we don't know for the patients still on OCS the dose they are taken.

The number of patients with SAHS is very small. I don't think SAHS can be used as biomarker based only on this observation.

There is no correlation between blood eosinophil levels before treatment and changes in OCS cycles, FEV1, number of exacerbations, atopy status in terms of allergen sensitivity. Upon this, there is a comment from the authors they "didn't evaluate the overall treatment response" which is not clear what it means. After all, its a real-life study.  

Author Response

(The authors gave the same response as above.)

Round 2

Reviewer 1 Report

Dear authors,

thank you for changes according to reccommendations.

Results and dicsussion are now better and easier to read.

Please include "flutisasone equivalent" also in the legends of tables 1-3.

I have no other reccommendations.

Thank you.

Regards

Reviewer 3 Report

The manuscript has improved with the changes. Nevertheless, there are still some aspects that need revision: 

- Methods, statistical analysis, regression: The information provided in the answer regarding the methodology used to reach the final multivariable model should be included in the manuscript. Even if you decided to keep in the model all variables that were significant, you should have assessed model accuracy and goodness of fit (it should be described in the methods and those results should be presented for each model). If you used linear regression in any of the models (line 443), you should specify which; it is not at all clear when was linear regression needed, considering the variables that you state as dependent, (which are binary; lines 431-439).   

- please check line 328 - "or hospitalization. The 73% (p=0.004) and 83% (p values not reported); 27% to 84% had discontinued maintenance OCS " and correct the sentence so that the reader can understand what 73% and 83% stand for.  
